# The Laro Kwo Project: A train the trainer model combined with mobile health technology for community health workers in Northern Uganda

Daniel Ebbs[1]☯*, Oyoo Benson[2]☯, Stanton Jasicki[3], Sarah McCollum[1], Michael Cappello[1,4]

1 Department of Pediatrics, Yale University School of Medicine, New Haven, Connecticut, United States of America, 2 Pader District Health Office, Pader District, Uganda, 3 Emergency Medical Associates, El Segundo, California, United States of America, 4 Department of Epidemiology of Microbial Diseases, Yale School of Public Health, New Haven, Connecticut, United States of America

☯ These authors contributed equally to this work.
* daniel.ebbs@yale.edu

**Data Availability Statement:** Data collected, VHT scores, are deidentified and deposited publicly on Open Science Framework. https://osf.io/mhd94/?

## Abstract

Community Health Workers (CHWs) in low and middle income countries (LMICs) provide invaluable health resources to their community members. Best practices for developing and sustaining CHW training programs in LMICs have yet to be defined using rigorous standards and measures of effectiveness. With the expansion of digital health to LMICs, few studies have evaluated the role of participatory methodologies combined with the use of mobile health (mHealth) for CHW training program development. We completed a three-year prospective observational study aligned with the development of a community-based participatory CHW training program in Northern Uganda. Twenty-five CHWs were initially trained using a community participatory training methodology combined with mHealth and a train-the-trainer model. Medical skill competency exams were evaluated after the initial training and annually thereafter to assess retention with use of mHealth. After three years, CHWs who advanced to trainer status redeveloped all program materials using a mHealth application and trained a new cohort of 25 CHWs. Implementation of this methodology coupled with longitudinal mHealth training demonstrated an improvement in medical skills over three years among the original cohort of CHWs. Further, we found that the train-the-trainer model with mHealth was highly effective, as the new cohort of 25 CHWs trained by the original CHWs exhibited higher scores when tested on medical skill competencies. The combination of mHealth and participatory methodologies can facilitate the sustainability of CHW training programs in LMIC. Further investigations should focus on comparing specific mHealth modalities for training and clinical outcomes using similar combined methodologies.

view_only=6fffe121df6a4172a12cc0d17f0f2b64
Ebbs, D. (2022, October 22). The Laro Kwo
Project: A train the trainer model combined with
mobile health technology for Community Health
Workers in Northern Uganda. Retrieved from osf.
io/mhd94.

**Funding:** Dr. Ebbs acknowledges support from
NIH T32: AI007210-39. The authors received no
other specific funding for this work.

**Competing interests:** The authors have declared
that no competing interests exist.

## Introduction

In remote regions of low-to-middle income countries (LMICs), access to healthcare is often limited, with significant disparity across socioeconomically disadvantaged communities. The added cost of health services for families living in extreme poverty is especially challenging when issues like food insecurity dominate day to day life. With inadequate health infrastructure and unaffordable quality healthcare, numerous alternative, community based models designed to build health capacity have been developed and implemented in recent decades [1–4].

Community health workers (CHWs) are local health workers who do not have a professional degree or license and who serve as health representatives within their communities. CHWs currently fill an extremely large gap in LMICs by providing health services and resources, including life-saving triage and treatment modalities to help connect sick patients from community to clinic [5–7]. Several studies have cited how CHWs are effective in meeting a diverse array of essential community health needs [1, 6]. However, with limited resources and training, significant challenges to providing structure and sustainable methodology for CHW program development exist. This, coupled with limited data on effective pedagogy for training CHWs in remote regions, has created a dire need for standardization to establish effective, sustainable, and community-rooted public health initiatives utilizing CHWs [7–9].

Northern Uganda is rural with limited health resources and poor socioeconomics, contributing to significant health disparities including the highest pediatric and maternal mortality rates in the country [10, 11]. Historically, decades of civil war across Northern Uganda displaced millions of people and delayed municipal growth and development, particularly limiting advancement in health infrastructure [12–14]. To assist in improving health disparities due to lack of health infrastructure and cyclic poverty, the Uganda Ministry of Health initiated a community health worker program in 2002, identifying community health workers in villages to construct Village Health Teams (VHT) [11–13]. With limited funding and support for VHT programs in Northern Uganda, no standardization exists for training, monitoring, or expanding these programs [11–13]. Given the potential for VHTs to reduce health disparities in rural areas, there is a critical need to support community health worker training programs with innovative methodology [11–14].

Program models employing methodologies that transition learner to instructor over a defined period of time have successfully demonstrated that this type of train-the-trainer methodology not only empowers those transitioning to instruct but ensures continued competency in learning objectives [2–4]. With this transitional model, sustainability is enhanced as the resource input is diminished, while local community leaders establish the infrastructure needed to develop local health capacity, including training and treatment sites [2–4]. The utility and sustainability of this model in limited health-resource settings rests heavily on cultural embeddedness with community-based interventions serving as a gateway to changes in health behavior and outcomes [2–4]. To ensure training success, a diversity of stakeholders is required and includes local community-based organizations, experts on education topics, community leaders, and elected or appointed officials [2–4].

Several new CHW training models have integrated the use of mobile health (mHealth) technology [15–17]. mHealth, a subset of electronic health (eHealth), consists of digital health information applied to mobile health technology [15–17]. With the widespread use of cellular phones, and more recently, tablet computers, information exchange across some of the most isolated, limited resource settings in the world is now feasible [18, 19]. With improved accessibility, the use of health information on mHealth can augment CHW training programs with improved training, monitoring, and treatment protocols [15–19]. Further, with limited available resources in many rural programs, the application of mHealth can provide recurrent

training and augment retention with repeated lessons in a setting without in-person instructors [15–19]. *Few studies to date have evaluated the effectiveness of combining train-the-trainer and mobile health methodologies for community-based health training programs in LMICs.* This study aims to address the effectiveness of combining a train-the-trainer model with mHealth by assessing medical skill competency over time and comparing initial trainee scores from two sets of instructors: trained CHWs and certified health professionals.

## Materials and methods

### Laro Kwo Project

The Laro Kwo Project (LKP) was founded in 2016 to establish a self-sustaining, participatory CHW training program led by community members in Northern Uganda. Laro Kwo, an Acholi name agreed upon by local villages in Northern Uganda, translates to 'saving lives' in English. The development of the LKP was facilitated by the nonprofit organizations Northern Uganda Medical Mission (NUMEM) and MGY (the name MGY represents the formula for potential energy in physics). NUMEM operates a health clinic in Pader, Northern Uganda, and MGY is a multidisciplinary team primarily composed of physicians and engineers based in the United States who provide resources for mobile health program development in limited resource settings.

The LKP was initiated after one year of discussions among village leaders, local elected officials, the Uganda Ministry of Health, and both NUMEM and MGY. Village leaders recommended a total of 25 CHWs among their communities to begin training under the LKP with an additional 25 added annually for the first two years of study. Preliminary meetings with the above stakeholders and community members identified specific health concerns that formed a framework for the training program curriculum. Once finalized, MGY drafted educational materials, including videos and quizzes of the requested training materials and presented them to stakeholders. Once approved by all stakeholders, NUMEM facilitated translations of materials and set up a date and time to begin the first training session with in-person didactics. The training curriculum focused on emergency triage, prehospital emergency care, and public health.

Seventy-five CHWs were recruited for the LKP over three years, January 2016–2019, and represent a diverse group of community members previously appointed by their respective communities, Uganda Ministry of Health and village leaders (Table 1). Specific requirements for appointment included residing among the village represented, literacy in local language, 18 years of age, interest in health, and an acknowledgement by village leaders as dependable and trustworthy. CHWs represented six different parishes initially, of which expanded to 15 parishes after three years. Selection of CHW parish occurred after discussions with stakeholders, considering proximity to primary site of training in Pader district due to accessibility for monitoring at the start of the program. Subsequent parishes selected expanded outward over three years. The average time working as a CHW was seven years and majority of CHWs had some exposure to mHealth through cell-phone use. No CHWs completed formal education degrees after grade school, however, all CHWs read and write Acholi. Most CHWs understand and can speak English as a second language; none are paid salary and majority of work is on volunteer basis.

**Table 1. VHT demographics.**

| | |
|---|---|
| **Age (mean)** | **44.2** |
| **Sex** (% Female) | 20 |
| **Years Experience** (mean) | 7.2 |
| **Exposure to mobile Health** (%) | 91 |

## Medical skill retention

In year 1 of the LKP, the 25 selected CHWs completed a five-day course which was held at the health ministry quarters in Kilak sub-county in Northern Uganda. Present for the training was a team of MGY physicians and engineers, NUMEM physicians and coordinators, ministry of health officials, elected local representatives and local village leaders. After completing didactic training on specified skills by a team of Uganda physicians (Table 2), each CHW completed a 1-on-1 skill competency exam the last day of training. Skill competencies were validated from expert consensus on basic medical skills and reviewed by a team of local Uganda physicians to ensure linguistic, cultural, and medical adequacy. Discussions between both U.S. and Uganda physician teams agreed upon each step of the medical competency skill checklist prior to implementation of training. The same instructor evaluated one skill for consistency. Skill competency exams had a maximum score of 42 points, or 100%.

Annual trainings were repeated the following two years with reassessment of skills after this initial training. During each annual training, a new cohort of 25 CHWs was added. After two years of passing >80% on skill competency, CHWs were offered promotion to advanced CHW and instructor status. After the second annual training session, a cohort of six CHWs chose to transition to instructors and revised the core training materials on the mHealth application. This included video demonstrations for each skill, which they proceeded to instruct at the next (year 2) annual training.

All CHWs enrolled in the LKP were concurrently volunteering as a CHW in their villages and had varying forms of past training through the health ministry. To mitigate this as a confounder, the LKP included only skills in the study that CHWs had no formal exposure to in the past. Furthermore, a selection bias was imposed as initial selection was based on proximity to resources and extended outward to more rural regions throughout the study.

A three-year prospective observational study was completed to evaluate changes in competency exam scores for specific medical skills. Sample size was calculated based on the availability of CHWs who could be trained and feasibility due to number of trainers. With the available sample, the within-group change required a minimum effect size (Cohen's d) of approximately 0.3 or higher and approximately 0.5 or higher between groups (original vs new CHWs). Our sample met these effect sizes. We hypothesized that with personal tablet computers loaded with a LKP tailored mobile health application, the original 25 CHWs would maintain annual medical skill competency scores without significant lapse in retention. Further, we hypothesized that by using a train-the-trainer model with mHealth, advanced CHWs could transition to instructor status and disseminate material equally effectively.

## Statistical analysis

CHWs' skill competency exam scores are presented as mean (95% CI). The original 25 CHW scores were modeled over time using mixed linear regression. The original CHW initial scores were compared with the initial scores from new CHWs who started in year 3 using Student's t-test.

**Table 2. Annual medical skillscompleted by CHWs.**

| Medical Skills (6 total) |
| --- |
| Assess Respirations |
| Assess Pulse |
| Manual blood pressure |
| Haemorrhage Control |
| Rapid Trauma Evaluation |
| Record Temperature |

### Ethics statement

Ethics approval for the study design, recruitment, and methods was obtained from A.T. Still University School of Osteopathic Medicine Institutional Review Board and the Uganda National Council for Science and Technology Ethics Committee. Written consent was obtained from each participant in study. During CHW recruitment, it was clarified that there was no risk in declining to participate in study.

## Results

Deidentified competency exam scores were collected at the end of the training and input to a password protected Excel file (Microsoft) by the principal investigator (D. Ebbs). Subsequent data analysis was completed using SAS 9.4. Modeled mean skill competency exam scores for the original CHWs were 80.3% (76.7%, 84.0%) for year 1, 90.6% (87.5%, 93.6%) for year 2, and 88.7% (86.8%, 90.7%) for year 3 (Table 3). There was evidence for a meaningful difference in scores by year in the overall comparison (p <0.0001) as well as for pairwise comparisons (Table 3), with an increase between year 1 and year 2 (p <0.0001) and between year 1 and year 3 (p <0.0001). Alpha of 0.05 was used for level of statistical significance. There was no evidence for a meaningful difference in scores between years 2 and 3 (p = 0.26).

Comparing the original CHW initial scores with initial scores from CHWs who started in year 3 ("new CHWs"), the new CHWs had a higher mean initial score (Table 4) of 90.7% (89.0%, 92.4%) than the original CHW mean score of 80.3% (76.7%, 84.0%; p<0.0001). The scores from year 3 represent scores from CHWs that were trained by advanced CHWs, completing the train-the-trainer cycle. Throughout the study period, two original CHWs were lost to follow-up during both years two and three. All other CHWs were present for each training session with a total of three cycles. There was no missing data for the seventy-five CHWs except competency scores for two original CHWs years two and three.

## Discussion

This study demonstrates that the use of train-the-trainer with mHealth is an effective model for sustainable community program development in LMICs. After reconstructing core medical education materials and transitioning over three years to instructor, trained CHWs effectively delivered the training to 25 new CHWs. The scores of new CHWs averaged higher than previous trainings implemented by local and foreign health professionals, with sustained high competency scores in years 2 and 3 as advanced CHWs assumed leadership of the program.

Although this study demonstrates a successful training model, we acknowledge there are many limitations, and future replication with a larger sample size is needed. mHealth literacy can correlate to voluntary education use and confound results if CHWs are not accessing education throughout the year. To recognize and address this early in the study, the training

**Table 3. Original CHW scores.**

| Effect | year | Estimate | Lower | Upper | P Value |
|--------|------|----------|-------|-------|---------|
| **year** | 1 | 0.8033 | 0.7669 | 0.8398 | - |
| **year** | 2 | 0.9056 | 0.8751 | 0.9360 | < .0001 |
| **year** | 3 | 0.8872 | 0.8678 | 0.9067 | < .0001 |

Average CHW competency scores for first training with years
2 and 3 representing trainings by advanced CHWs. P-values
represent difference between first year scores.

**Table 4. A comparison between original CHWs scores (year one) and new CHWs trained by advanced CHWs (year three).**

| | CHW group | | | |
|---|---|---|---|---|
| | new (N = 25) | original (N = 25) | Total (N = 50) | P Value |
| **Total score** | | | | |
| Mean (SD) | 38.10 (1.70) | 33.74 (3.83) | 35.92 (3.67) | <0.001*** |
| Median (IQR) | 38.0 (36.5–39.5) | 34.0 (31.0–37.0) | 36.8 (34.0–39.0) | <0.001*** |
| **Percentage** | | | | |
| Mean (SD) | 0.91 (0.04) | 0.80 (0.09) | 0.86 (0.09) | <0.001*** |
| Median (IQR) | 0.9 (0.9–0.9) | 0.8 (0.7–0.9) | 0.9 (0.8–0.9) | <0.001*** |

included sessions on tablet use. Furthermore, program coordinators through NUMEM were available and followed up with CHWs throughout the year to answer questions on application use and function, including capacity to charge tablets. Future studies are planned to monitor video usage in detail and expand on innovative forms of mHealth. Furthermore, to reduce selection bias, this study will be replicated with a larger sample size and matching of CHW villages based on proximity to resources. This approach should reduce selection bias and ensure adequate sample representation.

CHWs provide a critical resource to community members that can influence patient outcomes in LMICs. Particularly in regions such as Northern Uganda, where no regional hospital exists, careful triage of community members who may require hospitalization can be life-saving [10, 12, 13]. Critical to this role is the maintenance of basic skill competency for effective triage evaluation. Many studies recognize the effectiveness of CHWs in health programs but fail to recognize the need for sustainability and note limited evidence for best practices in training modality [7, 20]. A recent systematic review of CHW training programs focusing on LMICs highlights the scarcity of evidence for CHW training impact and the need for investigation of effective training modalities [7]. Further, this review emphasized the importance of community embeddedness to program success, and how increased motivation and confidence can lead to improved program services. The results from this study support the tenet that community focused training programs improve confidence and motivation, which link to program success.

Similarly, recent reviews of mHealth and interventions among CHWs outline the effectiveness of training CHWs with mHealth. However, these studies note several challenges, such as lack of culturally relevant mHealth interventions, lack of consistent methodology to assess outcomes, and need for further studies evaluating effective training modalities [7, 21, 22]. This study addresses some of these concerns and highlights how a community-based training program can sustain effective health training in limited resource settings. Moreover, and fundamental to this concept, the study demonstrates that combining train-the-trainer with mHealth ensures cultural relevance to education materials while driving motivation and self-empowerment. Future studies replicating this methodology will explore combinations of training modalities that are community-based and culturally tailored to increase sustainability. Additional studies addressing clinical outcomes of various CHW program models are necessary to validate a positive health impact of programs such as the LKP. As the Laro Kwo Project continues to expand in Northern Uganda, plans are underway to conduct a larger multi-village study using similar training methodology, with comparison of retention rates using different mHealth applications.

## Supporting information

**S1 File. Inclusivity questionnaire.**
(DOCX)

## Acknowledgments

The authors would like to acknowledge the time and commitment to training organized and implemented by the nonprofit organizations MGY and Northern Uganda Medical Mission; and the Uganda Ministry of Health. Furthermore, we would like to acknowledge David Tseng and Sam Waggoner for the development of a community tailored mobile health application. Finally, we would like to acknowledge the hard work and commitment of the 75 community health workers who selflessly provide critical health resources in service to their communities.

## Author Contributions

**Conceptualization:** Daniel Ebbs, Oyoo Benson, Stanton Jasicki.

**Data curation:** Daniel Ebbs.

**Formal analysis:** Sarah McCollum.

**Investigation:** Oyoo Benson, Stanton Jasicki.

**Methodology:** Daniel Ebbs.

**Project administration:** Daniel Ebbs, Oyoo Benson.

**Resources:** Daniel Ebbs, Oyoo Benson.

**Supervision:** Michael Cappello.

**Writing – original draft:** Daniel Ebbs, Oyoo Benson.

**Writing – review & editing:** Michael Cappello.

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
