## [Decision Letter · Decision Letter 0]

3 Jan 2023

PGPH-D-22-01696

The Laro Kwo Project: A train the trainer model combined with mobile health technology for Community Health Workers in Northern Uganda

Dear Dr. Ebbs,

Thank you for submitting your manuscript to PLOS Global Public Health. After careful consideration, we feel that it has merit but does not fully meet PLOS Global Public Health’s publication criteria as it currently stands. Therefore, we invite you to submit a revised version of the manuscript that addresses the points raised during the review process. Ensure to provide clarity on background and methods as indicated by the reviewers. 

We look forward to receiving your revised manuscript.

Kind regards,

Roopa Shivashankar, MD, MSc

Academic Editor

Journal Requirements:

2. Please send a completed 'Competing Interests' statement, including any COIs declared by your co-authors. If you have no competing interests to declare, please state "The authors have declared that no competing interests exist". Otherwise please declare all competing interests beginning with the statement "I have read the journal's policy and the authors of this manuscript have the following competing interests:"

Additional Editor Comments (if provided):

Reviewers' comments:

Reviewer's Responses to Questions

**Comments to the Author**

1. Does this manuscript meet PLOS Global Public Health’s publication criteria? Is the manuscript technically sound, and do the data support the conclusions? The manuscript must describe methodologically and ethically rigorous research with conclusions that are appropriately drawn based on the data presented.

Reviewer #1: Partly

Reviewer #2: Partly

2. Has the statistical analysis been performed appropriately and rigorously?

Reviewer #1: Yes

Reviewer #2: No

3. Have the authors made all data underlying the findings in their manuscript fully available (please refer to the Data Availability Statement at the start of the manuscript PDF file)?

Reviewer #1: No

Reviewer #2: No

4. Is the manuscript presented in an intelligible fashion and written in standard English?

Reviewer #1: Yes

Reviewer #2: Yes

5. Review Comments to the Author

Reviewer #1: Study title: The Laro Kwo Project: A train the trainer model combined with mobile health technology for community health workers in Northern Uganda.

Background:

The study is a three-year prospective observational study aligned with the development of a community-based participatory CHW training program in Northern Uganda. Twenty-five CHWs were initially trained using a community participatory training methodology combined with mHealth and a train-the-trainer model. Medical skill competency exams were evaluated after the initial training and annually thereafter to assess retention with use of mHealth. After three years, CHWs who advanced to trainer status redeveloped all program materials using a mHealth application and trained a new cohort of 25 CHWs.

Inputs:

1. The study is well articulated and presents the results comprehensively. The problem statement is clear and the research gap is identified.

2. The involvement of local elected representatives and local village leaders along with other officials for the conceptualisation of LKP and training is a refreshing strategy and a replicable example of communitization.

3. The introduction needs to give context of the Ugandan CHW program. Findings of desk review of CHW policy would be helpful to build the context for the study site.

4. A flow chart of the rollout of the intervention may be added for showcasing “Train-the-trainer” model adapted by the study. This will also clarify number of participants in the study. Exclusion and inclusion criteria (if any) may also be added.

5. The study may also need to include participant profile – age, gender, education qualification, experience in years/ months working as CHW, households/ population covered (if applicable) etc. Adding these details shall be helpful for understanding the study setting. Additionally, a baseline data of CHW/participant’s digital literacy could be mentioned, if gathered. This would also set a context of a pre-training for use of mobile applications.

6. The study does not mention the background qualification/skill set of the CHWs which could be an important confounding factor in assessing the retention of the training. Any other perceived confounding factors could also be mentioned.

7. Limitations of the study need to be mentioned. A section on scope for supplementing the initiative with further research could be mentioned.

8. The study does not mention whether it has adhered to STORBE guidelines. The study may reframe if it is deemed fit by the journal. (appraisal against STROBE checklist is attached as Annexure)

 

Annexure: Appraisal against STROBE checklist:

Item No Recommendation Present Article

Title and abstract 1 (a) Indicate the study’s design with a commonly used term in the title or the abstract Present

(b) Provide in the abstract an informative and balanced summary of what was done and what was found Present

Introduction

Background/rationale 2 Explain the scientific background and rationale for the investigation being reported Present

Objectives 3 State specific objectives, including any prespecified hypotheses Present

Methods

Study design 4 Present key elements of study design early in the paper Present

Setting 5 Describe the setting, locations, and relevant dates, including periods of recruitment, exposure, follow-up, and data collection Setting- the study area not explained, village name not mentioned.

Study period: not mentioned

Comparison: comparison was done with the new scores of the CHWs trained by the professional trainers and the mean score of new CHWs.

Data collection: the

Participants 6 (a) Cohort study—Give the eligibility criteria, and the sources and methods of selection of participants. Describe methods of follow-up

Case-control study—Give the eligibility criteria, and the sources and methods of case ascertainment and control selection. Give the rationale for the choice of cases and controls

Cross-sectional study—Give the eligibility criteria, and the sources and methods of selection of participants The eligibility criteria and selection of participants were not clearly defined, they were selected based on the nomination by the village leader.

The demographic details, qualification and past field of work experience was not explained.

(b) Cohort study—For matched studies, give matching criteria and number of exposed and unexposed

Case-control study—For matched studies, give matching criteria and the number of controls per case

Variables 7 Clearly define all outcomes, exposures, predictors, potential confounders, and effect modifiers. Give diagnostic criteria, if applicable Outcome: only the modelled mean skill competency was calculated.

Potential confounders: the confounders are not mentioned, the past field of experience and qualification of new CHWs can be confounder factor.

Data sources/ measurement 8* For each variable of interest, give sources of data and details of methods of assessment (measurement). Describe comparability of assessment methods if there is more than one group Primary data collection was done

Bias 9 Describe any efforts to address potential sources of bias The selection of new CHWs are not defined leading to selection bias

Study size 10 Explain how the study size was arrived at The sample size calculation not explained in the article

Quantitative variables 11 Explain how quantitative variables were handled in the analyses. If applicable, describe which groupings were chosen and why The quantitative variables were analysed in the study. Mean score was calculated for each of the CHWs based on skill competency exam. Skill competency exam was conducted on the 6 medical skills.

Statistical methods 12 (a) Describe all statistical methods, including those used to control for confounding No statistical methods used to address confounding factor

(b) Describe any methods used to examine subgroups and interactions Skill competency exam was done and mean and median score of new CHWs and old CHWs were compared

(d) Cohort study—If applicable, explain how loss to follow-up was addressed

Case-control study—If applicable, explain how matching of cases and controls was addressed

Cross-sectional study—If applicable, describe analytical methods taking account of sampling strategy No efforts taken to address the loss to follow up during the study period of 3 years.

(e) Describe any sensitivity analyses Not mentioned in the article

Continued on next page 

Results

Participants 13* (a) Report numbers of individuals at each stage of study—eg numbers potentially eligible, examined for eligibility, confirmed eligible, included in the study, completing follow-up, and analysed the details of participants were not explained. Only the number of participants selected were mentioned but eligibility, selection criteria and loss to follow-up not explained.

(b) Give reasons for non-participation at each stage Not explained

(c) Consider use of a flow diagram Not explained

Descriptive data 14* (a) Give characteristics of study participants (eg demographic, clinical, social) and information on exposures and potential confounders demographic, clinical, social characteristics of participants not explained

(b) Indicate number of participants with missing data for each variable of interest Not explained in the article

(c) Cohort study—Summarise follow-up time (eg, average and total amount) Follow up time is mentioned but number of number of participants lost are newly added to the study are not mentioned.

Outcome data 15* Cohort study—Report numbers of outcome events or summary measures over time The scores of CHWs for consecutive 3 years were mentioned

Main results 16 (a) Give unadjusted estimates and, if applicable, confounder-adjusted estimates and their precision (eg, 95% confidence interval). Make clear which confounders were adjusted for and why they were included Not done in the article

Other analyses 17 Report other analyses done—eg analyses of subgroups and interactions, and sensitivity analyses Mean, median analysed

Discussion

Key results 18 Summarise key results with reference to study objectives Done

Limitations 19 Discuss limitations of the study, taking into account sources of potential bias or imprecision. Discuss both direction and magnitude of any potential bias The limitations of the study can be sample size is small, the methodology if selection of CHWs is not clear, inconsistent methodology.

Interpretation 20 Give a cautious overall interpretation of results considering objectives, limitations, multiplicity of analyses, results from similar studies, and other relevant evidence Interpretation is done

Generalisability 21 Discuss the generalisability (external validity) of the study results No external validity done

Other information

Funding 22 Give the source of funding and the role of the funders for the present study and, if applicable, for the original study on which the present article is based No funding received

Reviewer #2: General Comments from the review

1. The structured IMRaD (Introduction, methods, results and discussion) format of abstract was absent in the manuscript.

2. Introduction:

It would be useful if the following information are also provided to clarify the context for better understanding of the global audience:

- Is there any specific name of the community health worker in Uganda?

- What is the population catered by each CHW in Uganda generally, i.e. is there a population norm for CHW.

- Since what time CHWs have been existing in the health system of Uganda?

- What are the minimum educational qualifications required to be a CHW, as this would determine the level of acceptance of mobile health technology?

Other comments -

- The sentence on line number 66 and 67 needs to be grammatically modified.

- What is train the trainers model as the rationale from various studies have mentioned about the utility and

the sustainability of the model?

- Who were the stakeholders required to run this train the trainer model apart from the community leaders and

CHWs as per the literature?

- References for the line number 76, 84 and 85 should be provided.

3. Methodology:

- Study period of the three year training is not mentioned unlike the project (2016)

- Why exactly 25 CHWs? Is there any specific rationale for including only so much? In other words, how was

sample size calculated?

- Sampling technique should be added.

- Who was the instructor for the didactic training among the team? This needs to mentioned.

- In evaluating skills for consistency by the instructor, was any skill out of 6 medical skills enlisted in Table 1

used for evaluation or was it a mere repetition of only one consistency skill by all 25?

- How was the evaluation performed through checklist? If it was through a checklist, then was it a validated

one?

- What was the duration of evaluation of each competency?

- Was 25 new CHWs enrolled every year as the initial cohort undergoes second and third year of training

leading to 75 CHWs in total? This number 75 has been mentioned only in the acknowledgement. This needs

clarification.

4. Statistical Analysis:

- How data entry was done and its quality was assured, has to be mentioned

- Was any software used for data analysis? If yes, then mention it.

- Level of statistical significance for P value considered by the author needs to be mentioned.

- Was there any data missing? If yes, then mention wherever applicable.

- Was there any loss to follow up in the subsequent years?

5. Results:

- Descriptive statistics of the cohort including socio-demographic information and years of experience as CHW

should be added if available.

- The outcome in the objective of comparing the initial trainee scores from the certified health professionals

apart from the trained CHWs was not found in the results section.

6. Discussion:

- Is too short and not all the points have been discussed. This needs to be rewritten.

- Limitations and strengths need to mentioned clearly.

Specific recommendations to be given apart from need of future studies.

6. PLOS authors have the option to publish the peer review history of their article (what does this mean?). If published, this will include your full peer review and any attached files.

**Do you want your identity to be public for this peer review?** For information about this choice, including consent withdrawal, please see our Privacy Policy.

Reviewer #1: No

Reviewer #2: No

---

## [Editor Report · Decision Letter 1]

22 Feb 2023

PGPH-D-22-01696R1

The Laro Kwo Project: A train the trainer model combined with mobile health technology for Community Health Workers in Northern Uganda

Dear Dr. Ebbs,

Thank you for submitting your manuscript to PLOS Global Public Health. This manuscript is being sent back to you as per your request to submit the revised version. 

We look forward to receiving your revised manuscript.

Kind regards,

Roopa Shivashankar, MD, MSc

Academic Editor
---

## [Decision Letter · Decision Letter 2]

21 Apr 2023

The Laro Kwo Project: A train the trainer model combined with mobile health technology for Community Health Workers in Northern Uganda

PGPH-D-22-01696R2

Dear Dr Ebbs,

We are pleased to inform you that your manuscript 'The Laro Kwo Project: A train the trainer model combined with mobile health technology for Community Health Workers in Northern Uganda' has been provisionally accepted for publication in PLOS Global Public Health.

Best regards,

Roopa Shivashankar, MD, MSc

Academic Editor

Reviewer Comments (if any, and for reference):

Reviewer's Responses to Questions

**Comments to the Author**

1. If the authors have adequately addressed your comments raised in a previous round of review and you feel that this manuscript is now acceptable for publication, you may indicate that here to bypass the “Comments to the Author” section, enter your conflict of interest statement in the “Confidential to Editor” section, and submit your "Accept" recommendation.

Reviewer #2: All comments have been addressed

2. Does this manuscript meet PLOS Global Public Health’s publication criteria? Is the manuscript technically sound, and do the data support the conclusions? The manuscript must describe methodologically and ethically rigorous research with conclusions that are appropriately drawn based on the data presented.

Reviewer #2: Partly

3. Has the statistical analysis been performed appropriately and rigorously?

Reviewer #2: Yes

4. Have the authors made all data underlying the findings in their manuscript fully available (please refer to the Data Availability Statement at the start of the manuscript PDF file)?

Reviewer #2: Yes

5. Is the manuscript presented in an intelligible fashion and written in standard English?

Reviewer #2: Yes

6. Review Comments to the Author

Reviewer #2: None

7. PLOS authors have the option to publish the peer review history of their article (what does this mean?). If published, this will include your full peer review and any attached files.

**Do you want your identity to be public for this peer review?** For information about this choice, including consent withdrawal, please see our Privacy Policy.

Reviewer #2: No
